# CY-Bench : A comprehensive benchmark dataset for sub-national crop yield forecasting

**D Paudel**[1] **H Baja**[1] **R van Bree**[1] **M Kallenberg**[1] **S Ofori-Ampofo**[2] **A Potze**[1] **P Poudel**[3]
**A Saleh**[4] **W Anderson**[5] **M von Bloh**[2] **A Castellano**[6] **O Ennaji**[7] **R Hamed**[8] **R Laudien**[9]
**D Lee**[10] **I Luna** [11] **D Masiliunas**[1] **M Meroni**[12] **S Mkuhlani** [13] **J Mutuku**[14] **J Richetti**[15]
**A Ruane**[6] **R Sahajpal** [5] **G Shuai**[5] **V Sitokonstantino**[11] **R de S. Nóia Jr**[2] **A Srivastava**[16]
**R Strong** [17] **L Sweet**[18] **P Vojnović**[12] **A de Wit** [1] **M Zachow**[2] **I Athanasiadis**[1]

[1]Wageningen Uni. & Research  [2]Technical Uni. Munich  [3]Purdue Uni.  [4]Ankara Uni.
[5]Uni. Maryland  [6]NASA  [7]Univ. Mohammed VI  [8]VU Amsterdam  [9]PIK  [10]Uni. Manitoba
[11]Uni. València  [12]JRC  [13]IITA  [14]ICRISAT  [15]CSIRO  [16]ZALF  [17]Texas Uni.  [18]UFZ

## Abstract

In-season or pre-harvest crop yield forecasts are essential for enhancing transparency in commodity markets and for planning towards achieving the United Nations' Sustainable Development Goal 2 of zero hunger, especially in the context of climate change and extreme events leading to crop failures. Pre-harvest crop yield forecasting is a difficult problem, as several interacting factors contribute to yield formation, including in-season weather variability, extreme events, long-term climate change, pests, diseases and farm management decisions. Machine learning methods provide ways to capture complex interactions among such predictors and crop yields. Prior research in agricultural applications, including crop yield forecasting, has primarily been case-study based, which makes it difficult to compare modeling approaches and measure progress. To address this gap, we introduce CY-Bench (Crop Yield Benchmark), a comprehensive dataset and benchmark to forecast crop yields. We standardized data source selection, preprocessing and spatio-temporal harmonization of public sub-national yield statistics with relevant predictors such as weather, soil, and remote sensing indicators, in collaboration with domain experts such as agronomists, climate scientists, and machine learning researchers. With CY-Bench we aim to: (i) standardize machine learning model evaluation in a framework that covers multiple farming systems in more than twenty-five countries across the globe, (ii) facilitate robust and reproducible model comparison through a benchmark addressing real-world operational needs, (iii) share a dataset with the machine learning community to facilitate research efforts related to time series forecasting, domain adaptation and online learning. The dataset and code used will be openly available, supporting the further development of advanced machine learning models for crop yield forecasting that can be used to aid decision-makers in improving global and regional food security.

**Keywords**: benchmark dataset; crop yield forecasts; agriculture; food security.

## 1 Introduction

Despite steady improvements in the efficiency of agricultural production over the last decades, the global food system is still rife with inequalities (60; 1), such as disproportionate access to resources

Submitted to the 38th Conference on Neural Information Processing Systems (NeurIPS 2024) Track on Datasets and Benchmarks. Do not distribute.

between developed and developing countries. The interconnectedness of countries and international trade can help to smooth swings in commodity prices, but can also bring intra-annual price volatility to import-dependent countries (81; 12; 80). Experts have emphasized the need for improved data, maps, and predictions (20; 37; 17). In particular, pre-harvest yield forecasts are vital for improving global market transparency and enabling decision-makers to plan response actions to mitigate anticipated shortages (65; 63; 7).

National and sub-national crop yield forecasts are produced by both private sector and governmental institutes using a combination of statistical modeling approaches and process-based crop models (5; 58; 23). Due to the multiplicity of systems and hazards involved, and the importance of compounding effects which are not yet well-understood, data-driven methods provide less explored ways to capture the complex and nonlinear relationships driving crop growth and development(59; 31). Additionally, the availability of high-quality agricultural data varies significantly by region and by crop; recent developments in transfer learning and domain adaptation may be useful for serving data-scarce regions or neglected and under-utilized crops. Over the recent years, several review articles (14; 25; 32; 73; 8; 42) and publications have highlighted excellent performance of machine learning for pre-harvest yield forecasting (79; 19; 28; 36; 44; 45; 76; 34). However, the data and code used in these studies are often unavailable, meaning that the results cannot be reproduced, and the diverse range of evaluation procedures, metrics, and datasets used in these studies means that synthesizing their results is difficult.

In order to better understand the specific strengths and weaknesses of existing machine learning methods for pre-harvest yield forecasting, and to drive further research progress, well-specified benchmark datasets compiled by domain experts are vital (53; 67; 16)(Sweet et al. in review). These benchmark datasets must reflect the needs of the worldwide community (41; 71). Recently, researchers have emphasized the need for machine learning benchmark datasets that include data from more regions and countries (50). Additionally, while forecast accuracy is crucial, machine learning models must also be reliable in settings comparable to real-world use in order to be adopted by stakeholders (72). The evaluation metrics used should closely represent the needs of stakeholders and allow a more granular breakdown of model performance (66; 11) - for example, the model's ability to capture yield variability in years with climate extremes must be reported (77). Finally, to avoid overestimation of model skill, the evaluation procedure must take into account the specific challenges arising from the use of non-i.i.d spatiotemporal data (40; 64; 26).

We present CY-Bench, a comprehensive dataset and benchmark for sub-national crop yield forecasting, with coverage of major crop-growing countries across the world for maize and wheat. Here, sub-national refers to the administrative levels for which official crop statistics are published; crop yield refers to the end-of-season yield reported in the statistics; and forecasting refers to the production of end-of-season yield estimates with a certain lead time before harvest (e.g. mid-season or 30 days before harvest) or before the publication of official statistics. Thus, the dataset combines sub-national yield statistics with relevant predictors, such as growing-season weather indicators, remote sensing indicators, evapotranspiration, soil moisture indicators, and static soil properties. CY-Bench has been designed and curated by agricultural experts, climate scientists, and machine learning researchers from the AgML community (`https://www.agml.org/`), with the aim of facilitating model intercomparison across the diverse agricultural systems around the globe in conditions as close as possible to real-world operationalization. Ultimately, by lowering the barrier to entry for ML researchers in this crucial application area, CY-Bench will facilitate the development of improved crop forecasting tools that can be used to support decision-makers in food security planning worldwide.

## 2 Related work

Crop yields are commonly forecast using weather, soil, moisture and crop productivity or remote-sensing-derived vegetation health indicators as predictors. Methods used include field surveys, process-based crop models, statistical regression and machine learning (5; 58). Data-driven approaches are appealing as they can capture processes not yet well-covered by biophysical crop

models, but typically require access to predictor data and yield data over large areas and spanning multiple years. The availability of these datasets determines the type of yield forecasting setup, which can range from national and sub-national level to field level. For example, the European Commission's Joint Research Centre (EC-JRC) regularly produces national crop yield forecasts for the EU and surrounding countries using crop models, agro-meteorological analyses and expertise of analysts (72). Sub-national yield forecasting utilizes data for a large number of sub-national administrative units (e.g. regions, provinces) typically collected by national statistical offices and captures spatial yield variability within a country (38; 44), which is crucial for targeted food security planning.

An increasing number of publications have demonstrated excellent performance of a diverse range of machine learning approaches for crop yield forecasting (34; 35; 49; 76; 45). Unfortunately, while results suggest that machine learning methods have great potential for providing accurate and timely crop yield forecasts, the datasets used by previous studies are, in most cases, unpublished. This has prevented the community from reproducing their results or comparing the strengths and weaknesses of different methods across different crops and regions. To our knowledge, SustainBench (78), which curates multi-source data for various tasks spanning the United Nations' seven sustainable development goals, includes a benchmark dataset designed to measure the performance of machine learning models for crop yield prediction. However, it targets end-of-season prediction for only one crop (soybean) in three countries (United States, Brazil and Argentina) and uses a relatively small set of predictors. Another public dataset is CropNet (33), which only includes the United States. Similarly, there are ongoing efforts (82) to produce a multi-task benchmark dataset which includes yield prediction in the USA as a sub-task. Apart from these, other available data contributions include yield statistics only (15; 48; 47; 54; 3; 4; 10; 13; 24; 39) or have been made available in combination with predictor data published with existing studies (28; 21; 43; 45) but are not explicitly tailored for yield forecasting benchmarking studies.

In comparison, CY-Bench data covers forty-two countries across six continents. This enables a comprehensive evaluation of model performance across regions with heterogeneous agricultural practices and infrastructure, including developing countries which are generally under-represented in machine learning benchmarks. Furthermore, we closely mimic real-world operationalization settings in the predictor data used, data pre-processing steps and evaluation set-up, including the use of temporal Leave-One-Year-Out validation (as opposed to the random sampling methods used in SustainBench and multiple previous studies). This means that novel machine learning methods which achieve excellent performance on CY-Bench could be used to improve yield forecasting systems in practice, providing accurate and timely information urgently needed by decision-makers.

Although we have identified a distinct lack of benchmark datasets for agricultural yield forecasting, there have been many recent developments in the related field of crop type mapping using satellite data (55; 69; 78; 29), leading to exciting progress in the development of methods for extracting meaningful patterns from time series of earth observation data (56; 55; 46; 57). Other related work (70; 68; 27) has been able to exploit meta-learning and multitask learning to improve model performance for land cover classification, crop mapping and agricultural yield forecasting. While CY-Bench is focused on pre-harvest yield forecasting, the dataset includes time series of crop productivity or vegetation health indicators from earth observation as predictors, and can therefore be easily combined with existing crop mapping benchmark datasets to explore such approaches.

## 3 CY-Bench task and datasets

### 3.1 Task

CY-Bench is designed to evaluate model performance for in-season crop yield forecasting at sub-national level. Forecasts are generated for selected crops (maize and wheat) at different time points, based on stakeholder needs (e.g. mid-season, a quarter of the season, or a certain number of days before harvest). For this exercise, we only report forecasts generated mid-season, the timing of which can differ by location. Mid-season was selected because peak model performance is typically

reached around the mid-point of the growing season. This mid-point is also when the transition from vegetative to reproductive growth stage happens for most crops (30; 5). Season length and mid-season information is derived from crop calendars. As in the operational setting, models must forecast the end-of-season crop yield outcomes based on the available time series data only up until the designed lead time.

## 3.2 Dataset overview

**Agricultural yield data.** The CY-Bench dataset includes crop statistics from twenty-nine countries for wheat and forty-two countries for maize (see Figure 1, 2). Models are trained to predict official crop yield statistics for sub-national administrative levels, which are obtained from national statistics offices (e.g. National Agricultural Statistics Service of the United States Department of Agriculture) or regional agencies (e.g. Eurostat and FEWSNET). Details of each source are indicated in the data preparation section in GitHub. Depending on the country, the term 'sub-national' can refer to administrative division 1 (province, state, region), division 2 (district), or division 3 (county, municipality, commune). When statistics for multiple administrative levels are available, we select the highest resolution.

**Predictor data.** CY-Bench predictor data includes static soil properties and time series of weather variables, soil moisture indicators and vegetation indicators (Table 1). Soil data comes from the WISE Soil database (6), weather variables from AgERA5 (9), potential or reference evapotranspiration (ET0) from FAO-AQUASTAT (2), soil moisture indicators from GLDAS (52), vegetation indicators (fraction of absorbed photosynthetically active radiation (FPAR) and normalized difference vegetation index (NDVI)) from EC-JRC and MODIS MOD09CMG respectively (62; 75). Predictor data and yield statistics often differ in spatial and temporal resolution, requiring further processing to align them effectively. Weather, ET0 and soil moisture data come in daily time steps. FPAR comes in dekadal time step, with three values per month (days 1-10, 11-20, 21-31). NDVI data is available approximately every week, but the dates are not regular. Predictor data is filtered using crop-type maps (or crop masks) from EC-JRC (18), which are derived from the WorldCereal project (74). This step restricts predictor data to pixels in harvested crop areas only. When crop masks and predictor data differ in resolution, the crop mask is resampled to the resolution of the predictor data. After masking, predictor data is aggregated to match the boundaries and spatial level of the yield data according to the administrative level (Figure 3). Additionally, as the sensitivity of crops vary throughout the phenological cycle, time series predictor data must correspond to the growing season. As this depends on the specific crop, management practices, and location, crop calendar information such as the start and end of season is required. In CY-Bench, these crop calendars are obtained from the WorldCereal project (22).

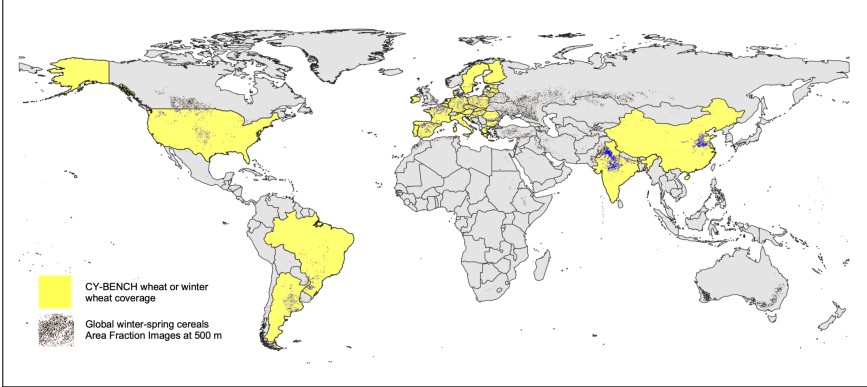

Figure 1: A map of the countries covered by CY-Bench for wheat yield forecasting. CY-Bench has coverage in 31 countries in total.

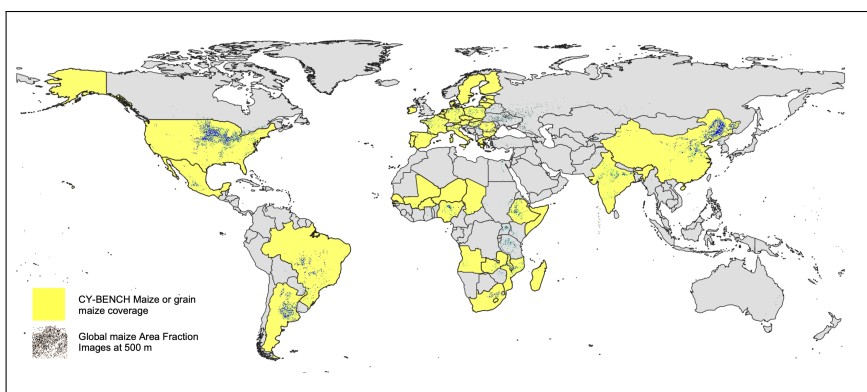

Figure 2: A map of countries covered by CY-Bench for maize yield forecasting. CY-Bench has coverage in 42 countries in total.

Table 1: Overview of the predictor data, crop mask and crop calendar. NDVI refers to the normalized difference vegetation index, FPAR is the fraction of absorbed photosynthetically active radiation and AWC is the available water capacity.

| Category | Data | | Spatial resolution | Temporal resolution | Source |
|---|---|---|---|---|---|
| | Name | Unit | | | |
| Meteorological | temperature precipitation solar radiation | °C mm $Jm^{-2}$ | 0.1° | daily | AgERA5 (9) |
| | evapotranspiration | mm | 0.1° | daily | AQUASTAT-FAO (2) |
| Vegetation | FPAR | % | 500m | 10-days | JRC (62) |
| | NDVI | - | 5000m | 8-days | MOD09CMG (75) |
| Soil | AWC bulk density drainage class | $cm\,m^{-1}$ $kg\,dm^{-3}$ - | 30" | static | WISE (6) |
| | moisture content | $kg\,m^{-2}$ | 0.25° | daily | NASA GLDAS (52) |
| Crop | crop mask crop calendar | - | 0.5° | - | Crop masks (74; 18) Crop calendars (22) |

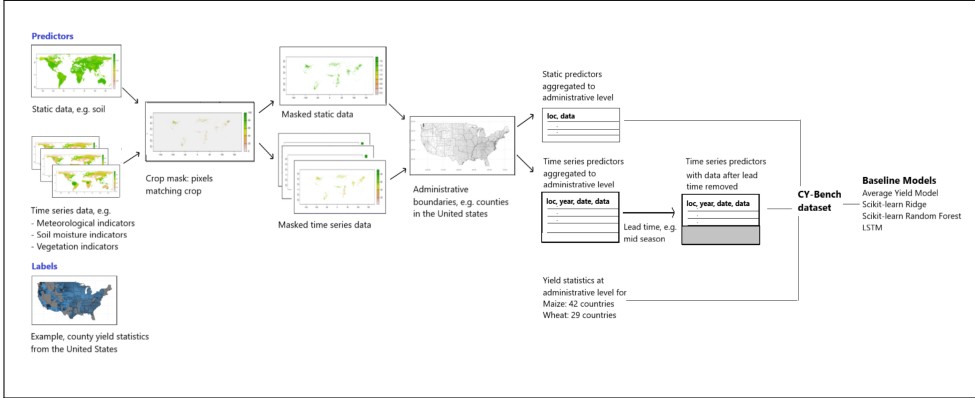

Figure 3: Overview of the CY-Bench data preparation process.

For deep learning models, such as Long Short Term Memory networks (LSTM), time series data is aggregated to dekadal time steps (days 1-10, 11-20, 21-31, and so on), which allows all datapoints to have the same number of time steps and therefore fixed input dimension.

For tree-based models and other machine learning models which are designed for tabular data, the time series data is aggregated in the temporal dimension to create domain-relevant features. These include monthly averages of minimum daily temperature ($tmin$), maximum daily temperature ($tmax$), average daily temperature, daily precipitation ($prec$), cumulative climatic water balance (prec - ET0) and surface soil moisture. Similarly, monthly maximum values were calculated for cumulative growing degree days ($GDD$), cumulative precipitation, cumulative FPAR and cumulative NDVI. Furthermore, we calculated the number of days in which $tmin$ was less than 0 degree Celsius ('cold days'), days in which $tmax$ was greater than 35 degrees Celsius ('hot days') and days where $prec$ was less than 1 mm ('dry days').

**Dataset access.** The dataset is available in Google Drive. A python library has been developed to load the datasets and run CY-Bench.

Table 2: Maize NRMSE per model for Argentina (AR), Brazil (BR), China (CN), Germany (DE), France (FR) and the United States (US).

| Model | AR | BR | CN | DE | FR | US |
|-------|----|----|----|----|----|----|
| LSTM | 87.206 | 42.352 | 20.584 | 13.778 | 21.967 | 23.962 |
| Naive | **33.514** | **33.284** | **9.384** | 14.838 | **16.860** | **18.101** |
| RF | 49.544 | 45.538 | 13.767 | **14.227** | 18.549 | 19.391 |
| Ridge | 152.41 | 64.363 | 48.245 | 48.798 | 23.043 | 22.443 |

Table 3: Wheat NRMSE per model for Argentina (AR), Brazil (BR), China (CN), Germany (DE), France (FR) and the United States (US).

| Model | AR | BR | CN | DE | FR | US |
|-------|----|----|----|----|----|----|
| LSTM | 36.440 | 33.137 | 99.883 | 14.782 | 17.540 | 35.700 |
| Naive | **24.349** | **28.008** | **10.808** | **9.941** | **9.546** | **19.410** |
| RF | 32.941 | 31.059 | 45.804 | 15.490 | 17.323 | 39.305 |
| Ridge | 31.061 | 31.737 | 351.01 | 60.968 | 65.382 | 29.093 |

## 4 Model evaluation and baselines

In CY-Bench, models are trained per country and per crop, and evaluated using Leave One Year Out (LOYO) evaluation. The motivation for LOYO is to obtain a robust estimate of the performance of algorithms on both average and extreme harvest years. As each season can vary substantially from previous years, measurement of predictive performance on only the current season or the most recent year may under- or over-estimate the forecasting ability of a model. For more information regarding model evaluation strategies in the context of agriculture see ((51)).

We evaluate the performance of four baseline models. First, the Average Yield model (*Naive*) predicts the average of the training set by administrative region (if present in training data) or country (if absent in training data). Second, the Ridge model (implemented in Scikit-Learn) builds a linear model using features designed as described in the previous sub-section. Third, Random Forest is used (also implemented in Scikit-Learn), which is frequently used for agricultural machine learning studies. Finally, we include LSTM as a baseline for representation learning from time series data.

As our evaluation metrics, we use the normalized root mean squared error (NRMSE; i.e., the root mean squared error normalized by the average yield of the test set), and mean absolute percentage error (MAPE). NRMSE and MAPE are reported by averaging over all cross-validation test folds (which covers the complete dataset for LOYO) and all admin regions with a country. Additionally, metrics and box plots describing model performance for each year individually are included in the Supplement.

We report results of the baseline model benchmarks in figures 4 and 5 for maize and wheat, respectively, to show NRMSE of different countries and baseline models. Moreover, we report median NRMSE of select countries for each model in tables 2 and 3 for maize and wheat, respectively.

The results show that the *Naive* model outperforms all the other baseline models, except for Random Forests. The *Naive* model is a test of prediction skill. The performance of most machine learning models shows the difficulty of generalizing from the training set.

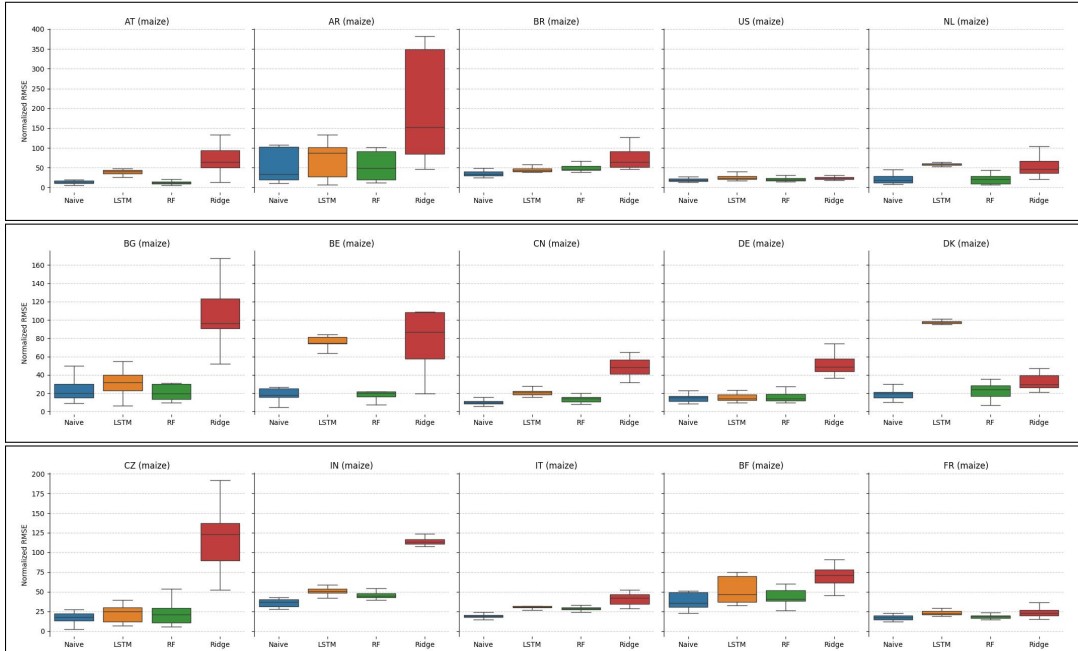

Figure 4: NRMSE for maize, predicted at mid-season lead time.

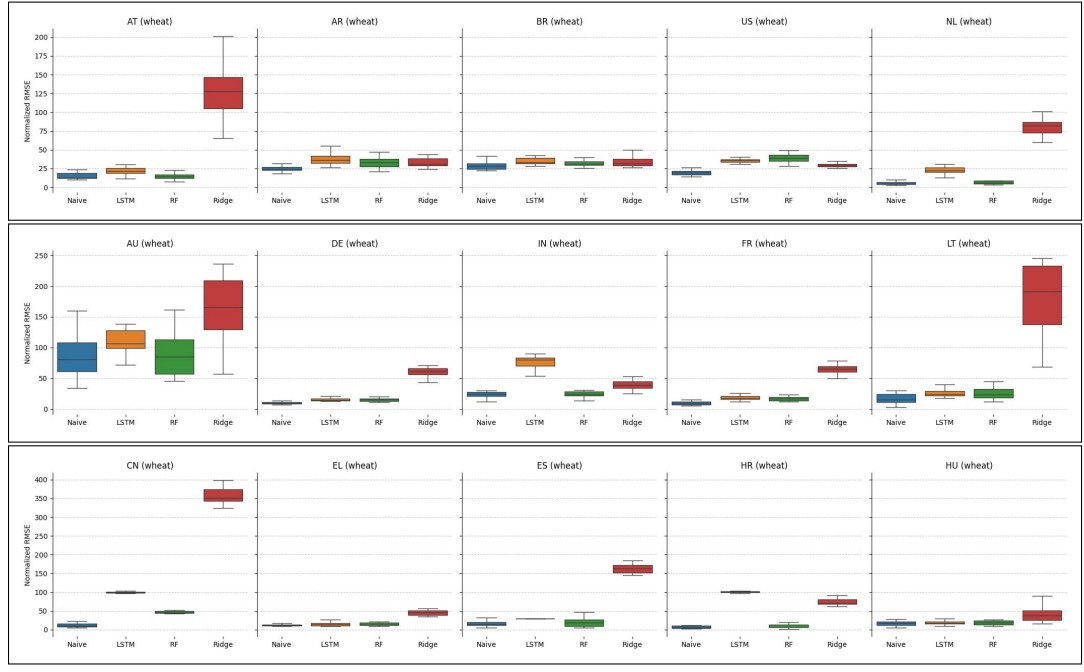

Figure 5: NRMSE for wheat, predicted at mid-season lead time.

# 5   Contributions, limitations and future work

In addition to the relevance for climate change, food security and United Nations' sustainable development goals, CY-Bench dataset is relevant to the ML community due to its comprehensive geographic coverage, capturing diverse agricultural practices and conditions. The inclusion of high-resolution satellite imagery, weather data, and soil properties provides a rich, heterogeneous dataset that presents numerous opportunities for the development of innovative machine learning methods. An inherent challenge of agricultural data, and crop-yield forecasting specifically, is the difficult and high level of domain knowledge required in collecting and processing the various data types and defining the task. This analysis-ready dataset is accessible to ML modelers who do not necessarily have to be experts in yield forecasting, lowering the barrier to entry for advanced yield forecasting research and fostering broader participation and innovation in the field. Beyond academic research, this dataset can significantly impact policy-making, agricultural planning, and disaster response by enabling the robust evaluation and development of operationalizable models. Researchers, policymakers, farmers, and agribusinesses can all benefit from the insights derived from this dataset, leading to better-informed decisions and improved agricultural outcomes.

Apart from the downstream task of in-season yield forecasting, CY-Bench enables explorations in transfer learning, domain adaptation, and representation learning. Researchers can leverage this dataset to assess if models are able to generalize well across diverse geographic and climatic conditions. While in this paper we focus on forecasting crop yields by training individual models for each crop and country, the dataset allows for a more integrated approach. We envision at least four directions for future research. First, transfer learning methods can be explored to improve model generalisation ability when training models on data from a data-rich region and deploying the forecasting model to data-sparse regions. Second, self-supervised learning could be used to harness the vast amounts of unlabeled agricultural data available. By training models to recognize patterns and structures within this data, we can build robust representations that capture essential features of the agricultural system. These representations can then be fine-tuned using the labeled datasets in CY-Bench specific to each country or crop. For instance, a self-supervised model trained on satellite images and environmental data can later be fine-tuned to predict specific crop yields in various regions, making it a powerful tool for global agricultural analysis. Third, another important area is to explore the stability of model predictions against natural and human interventions. This involves understanding how factors like extreme weather events, policy changes, or management practices impact yield forecasts. Causal invariant learning focuses on identifying and utilizing stable variables across different environments to ensure robustness and generalization. For example, soil quality and basic climatic factors like temperature and precipitation may have stable relationships with crop yields. By recognizing variables that consistently impact crop yields regardless of geographic or climatic differences, it may be possible to build models that are resilient to distributional shifts and perform reliably across diverse conditions. Fourth, deep learning techniques, such as autoencoders, can be employed to learn compact and informative representations of the input data, potentially uncovering latent variables that are more directly related to crop yields. This could improve the model's ability to generalize and perform well across different regions and conditions, while possibly giving scientific insight into the underlying drivers of agricultural crop yields.

We would like to also highlight several limitations and areas for improvement in future iterations of CY-Bench. First, some limitations stem from the data sources selected. The predictors do not capture certain factors that influence end-of-season yields, such as pests, diseases and farm management choices. Similarly, CY-Bench does not include socioeconomic factors such as market prices, labor availability, and policy changes. Including these variables could provide a more holistic understanding of yield fluctuations and help in developing more robust models. Additionally, our modeling setup does not differentiate between irrigated and non-irrigated systems. These systems can exhibit different responses to predictors due to varying water availability, leading to potential inaccuracies in yield forecasts. Our choice was driven by the fact that crop statistics in most countries are rarely reported separately for irrigated and non-irrigated areas. Second, CY-Bench does not supply process-based crop model outputs, which could be used as inputs to machine learning models, and features are

aggregated in fixed time steps, rather than being designed according to the stage of crop growth and development. Access to crop model outputs could provide information on key phenological state changes, which can be useful to design more predictive features. Third, crop yield forecasting models could benefit from incorporating weather forecasts. In our setup, models cannot access data after the lead time and, therefore, cannot capture conditions that might affect the end-of-season yields after that point. In the real-life setting, forecasters would have access to weather forecasts that may provide useful information. Finally, the LOYO method of evaluation is used due to small data sizes in many countries. This approach assumes that all years are independent, which may be too strong of an assumption if consecutive years have correlated environmental and climatic conditions.

## 6 Conclusion

Innovative data-driven solutions will be crucial to achieve the United Nations' Sustainable Development Goal 2 of Zero Hunger (61). By providing consistent evaluation of large-scale crop yield forecasts, CY-Bench is a step forward in bridging the gap between agricultural modeling and machine learning community. Curated by an interdisciplinary group of experts in agronomy, food security, climate science and agriculture, this dataset can facilitate increased collaboration between fields, and ultimately help to produce reliable crop yield forecasts to support decisions of farmers, policymakers and commodity traders worldwide.

## Acknowledgements

CY-Bench benefited from many helpful discussions with the participants of AgML, the Machine Learning team of the Agricultural Model Intercomparison and Improvement Project (AgMIP). We would also like to acknowledge the contributions of Marc Russwurm, Afef Marzougui, Hendrik Boogaard, Marijn van der Velde, Steven Hoek, Filip Szabo, Francesco Collivignarelli, Xiaomao Lin, Toshi Iizumi, Peng Fu, Paresh Shirsath, Soora Naresh Kumar, Sibiri Traore and Javier Garcia Navarro in the design and implementation of the benchmark and preparation of the manuscript.

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
