# Supplementary information for CY-Bench : A comprehensive benchmark dataset for sub-national crop yield forecasting

**D Paudel**[1] **H Baja**[1] **R van Bree**[1] **M Kallenberg**[1] **S Ofori-Ampofo**[2] **A Potze**[1] **P Poudel**[3]
**A Saleh**[4] **W Anderson**[5] **M von Bloh**[2] **A Castellano**[6] **O Ennaji**[7] **R Hamed**[8] **R Laudien**[9]
**D Lee**[10] **I Luna** [11] **D Masiliunas**[1] **M Meroni**[12] **S Mkuhlani** [13] **J Mutuku**[14] **J Richetti**[15]
**A Ruane**[6] **R Sahajpal** [5] **G Shuai**[5] **V Sitokonstantinou**[11] **R de S. Nóia Jr**[2] **A Srivastava**[16]
**R Strong** [17] **L Sweet**[18] **P Vojnović**[12] **A de Wit** [1] **M Zachow**[2] **I Athanasiadis**[1]
[1]Wageningen Uni. & Research  [2]Technical Uni. Munich  [3]Purdue Uni. [4]Ankara Uni.
[5]Uni. Maryland  [6]NASA  [7]Univ. Mohammed VI  [8]VU Amsterdam  [9]PIK [10]Uni. Manitoba
[11]Uni. València  [12]JRC  [13]IITA [14]ICRISAT  [15]CSIRO  [16]ZALF  [17]Texas Uni.  [18]UFZ

# 1 Dataset

## 1.1 Code Access and Use

**Access** The complete codebase encompassing data preprocessing, model construction, training, evaluation, and data/metric visualization routines can be accessed through out publicly accessible GitHub repository: `https://github.com/BigDataWUR/AgML-CY-Bench/`. A summarizing overview can be found on `https://cybench.agml.org/`. We additionally provide a Python package `cybench` that can be installed via the repository.

**Documentation** The dataset is available in Google Drive. The dataset is comprehensively documented using the framework of Data Cards (8). Each individual dataset subset is accompanied by a dedicated Data Card located within the data_preparation directory of our repository.

**Intended use** The CY-Bench dataset offers a valuable resource for the machine learning community, especially those interested in applying their expertise to real-world challenges in agriculture. Users can load the dataset and train their own models, and then evaluate using a standardized scheme, tailored for the context of agriculture.

**Contributing** AgML, the Machine Learning team of the Agricultural Model Intercomparison and Improvement Project, will be responsible for maintaining the dataset. To facilitate the dataset's ongoing growth, we've carefully crafted a process for incorporating new data or on-boarding additional datasets into our benchmark.

**Licensing** We, the authors, take full responsibility for any violations of intellectual property rights or other legal rights arising from our inclusion of data within this work. We have made our best effort to ensure all materials are properly attributed or used in accordance with their licenses. CY-Bench dataset is licensed under EUPL-1.2, which is compatible with all of the licenses for the datasets included. The manuscript itself is licensed under CC BY 4.0.

Submitted to the 38th Conference on Neural Information Processing Systems (NeurIPS 2024) Track on Datasets and Benchmarks. Do not distribute.

## 1.2 Dataset overview

**Crops** CY-Bench covers two main crops, namely maize and wheat. Depending on the country, the crop names can refer to different varieties or seasons of maize and wheat. Table 1 describes the representative crop name as provided in the country-specific crop statistics. Table 2 specifies the countries under groups EU and FEWSNET.

Table 1: Defining crop names as presented in crop statistics

| Country/Region | Admin level name | Maize | Wheat |
|---|---|---|---|
| EU-EUROSTAT | Admin level 2 or 3 | grain maize | soft wheat |
| Africa-FEWSNET | Admin level 1 or 2 | maize | NA |
| Argentina (AR) | department | corn | wheat |
| Australia (AU) | sub-state | NA | winter wheat |
| Brazil (BR) | municipality | grain corn | grain wheat |
| China (CN) | province | grain corn | winter wheat |
| Germany (DE) | district | grain maize | winter wheat |
| India (IN) | district | maize | wheat |
| Mali (ML) | municipality | maize | NA |
| Mexico (MX) | state | white/yellow corn | NA |
| United States (US) | county | grain corn | winter wheat |

Table 2: List of countries under groups EU and FEWSNET

| Group | Country name (country code) : Admin level | | |
|---|---|---|---|
| EU (n=22) | Austria (AT) : 2 | Belgium (BE) : 2 | Bulgaria (BG) : 2 |
| | Czech Republic (CZ) : 3 | Denmark (DK) : 3 | Estonia (EE) : 3 |
| | Greece (EL) : 3 | Spain (ES) : 3 | Finland (FI) : 3 |
| | France (FR) : 3 | Croatia (HR) : 2 | Hungary (HU) : 3 |
| | Ireland (IE) : 2 | Italy (IT) : 3 | Lithuania (LT) : 3 |
| | Latvia (LV) : 3 | Netherlands (NL) : 2 | Poland (PL) : 2 |
| | Portugal (PT) : 2 | Romania (RO) : 3 | Sweden (SE) : 3 |
| | Slovakia (SK) : 3 | | |
| FEWSNET (n=12) | Angola (AO) : 1 | Burkina Faso (BF) : 2 | Ethiopia (ET) : 2 |
| | Lesotho (LS) : 1 | Madagascar (MG) : 2 | Malawi (MW) : 2 |
| | Mozambique (MZ) : 1 | Niger (NE) : 2 | Senegal (SN) : 2 |
| | Chad (TD) : 1 | South Africa (ZA) : 1 | Zambia (ZM) : 2 |

## 1.3 Data source selection

Our selection of data sources was guided by the recommendations of a panel of experts. A detailed justification for our choices, including a discussion of alternative options considered, is available on `https://github.com/BigDataWUR/AgML-CY-Bench/blob/main/data_preparation/DATA-SOURCES-SELECTION.md`.

## 1.4 Data preparation

The script that implements the preparation workflow as outlined in section *3.2 Predictor data* and summarized in *Figure 3* of the main paper is given in predictor_data_prep.r

Table 3: Overview of dataset sizes for maize. NOTE: Yield data is available for most countries earlier than 2003. 2003 cutoff is due to soil moisture data starting from 2003. Some countries (e.g. EE, FI, IE, LV, MX) have no data after aligning with predictor data.

| SN | Crop, Country | Nr. years | Min year, max year | Nr. admin regions | Nr. data samples |
|----|---------------|-----------|--------------------|-------------------|------------------|
| 1 | maize, AO | 15 | 2003, 2017 | 18 | 240 |
| 2 | maize, AR | 21 | 2003, 2023 | 317 | 5564 |
| 3 | maize, AT | 18 | 2003, 2020 | 9 | 162 |
| 4 | maize, BE | 10 | 2011, 2020 | 11 | 85 |
| 5 | maize, BF | 16 | 2003, 2019 | 45 | 540 |
| 6 | maize, BG | 18 | 2003, 2020 | 6 | 108 |
| 7 | maize, BR | 21 | 2003, 2023 | 4017 | 70691 |
| 8 | maize, CN | 20 | 2003, 2022 | 31 | 610 |
| 9 | maize, CZ | 16 | 2005, 2020 | 14 | 212 |
| 10 | maize, DE | 19 | 2003, 2021 | 280 | 3362 |
| 11 | maize, DK | 10 | 2011, 2020 | 7 | 22 |
| 12 | maize, EE | 0 | NA | 0 | 0 |
| 13 | maize, EL | 11 | 2009, 2019 | 40 | 440 |
| 14 | maize, ES | 18 | 2003, 2020 | 47 | 830 |
| 15 | maize, ET | 14 | 2003, 2020 | 68 | 774 |
| 16 | maize, FI | 0 | NA | 0 | 0 |
| 17 | maize, FR | 18 | 2003, 2020 | 92 | 1628 |
| 18 | maize, HR | 16 | 2005, 2020 | 2 | 32 |
| 19 | maize, HU | 18 | 2003, 2020 | 20 | 359 |
| 20 | maize, IE | 0 | NA | 0 | 0 |
| 21 | maize, IN | 15 | 2003, 2017 | 543 | 6605 |
| 22 | maize, IT | 18 | 2003, 2020 | 102 | 1691 |
| 23 | maize, LS | 18 | 2004, 2021 | 10 | 163 |
| 24 | maize, LT | 18 | 2003, 2020 | 10 | 150 |
| 25 | maize, LV | 0 | NA | 0 | 0 |
| 26 | maize, MG | 6 | 2005, 2010 | 22 | 132 |
| 27 | maize, ML | 15 | 2003, 2017 | 24 | 360 |
| 28 | maize, MW | 6 | 2018, 2023 | 4 | 16 |
| 29 | maize, MX | 0 | NA | 0 | 0 |
| 30 | maize, MZ | 17 | 2004, 2022 | 10 | 159 |
| 31 | maize, NE | 17 | 2003, 2021 | 26 | 244 |
| 32 | maize, NL | 13 | 2008, 2020 | 12 | 126 |
| 33 | maize, PL | 18 | 2003, 2020 | 17 | 301 |
| 34 | maize, PT | 18 | 2003, 2020 | 5 | 90 |
| 35 | maize, RO | 18 | 2003, 2020 | 42 | 736 |
| 36 | maize, SE | 10 | 2007, 2020 | 1 | 10 |
| 37 | maize, SK | 12 | 2007, 2018 | 8 | 94 |
| 38 | maize, SN | 13 | 2003,2015 | 40 | 401 |
| 39 | maize, TD | 15 | 2003, 2017 | 17 | 231 |
| 40 | maize, US | 20 | 2003, 2022 | 2223 | 32510 |
| 41 | maize, ZA | 19 | 2004, 2022 | 9 | 167 |
| 42 | maize, ZM | 14 | 2004, 2017 | 71 | 994 |

## 2 Supplementary results

For figures and data visualizations we kindly refer to the results folder of our GitHub repository.

Table 4: Overview of dataset size for wheat. NOTE: Yield data is available for most countries earlier than 2003. 2003 cutoff is due to soil moisture data starting from 2003.

| SN | Crop, Country | Nr. years | Min year, max year | Nr. admin regions | Nr. data samples |
|---|---|---|---|---|---|
| 1 | wheat, AR | 21 | 2003,2023 | 266 | 4429 |
| 2 | wheat, AT | 18 | 2003,2020 | 9 | 135 |
| 3 | wheat, AU | 20 | 2003,2022 | 20 | 270 |
| 4 | wheat, BE | 18 | 2003,2020 | 11 | 183 |
| 5 | wheat, BG | 10 | 2010,2020 | 6 | 45 |
| 6 | wheat, BR | 20 | 2003,2022 | 1296 | 18498 |
| 7 | wheat, CN | 20 | 2003,2022 | 26 | 473 |
| 8 | wheat, CZ | 18 | 2003,2020 | 14 | 232 |
| 9 | wheat, DE | 19 | 2003,2021 | 365 | 5751 |
| 10 | wheat, DK | 14 | 2006,2020 | 10 | 136 |
| 11 | wheat, EE | 11 | 2005,2020 | 5 | 45 |
| 12 | wheat, EL | 17 | 2003,2019 | 43 | 629 |
| 13 | wheat, ES | 18 | 2003,2020 | 46 | 557 |
| 14 | wheat, FI | 17 | 2004,2020 | 18 | 93 |
| 15 | wheat, FR | 18 | 2003,2020 | 89 | 1524 |
| 16 | wheat, HR | 13 | 2008,2020 | 2 | 25 |
| 17 | wheat, HU | 18 | 2003,2020 | 20 | 297 |
| 18 | wheat, IE | 5 | 2010,2020 | 3 | 11 |
| 19 | wheat, IN | 15 | 2003,2017 | 494 | 6168 |
| 20 | wheat, IT | 18 | 2003,2020 | 94 | 1252 |
| 21 | wheat, LT | 18 | 2003,2020 | 10 | 144 |
| 22 | wheat, LV | 16 | 2003,2018 | 5 | 57 |
| 23 | wheat, NL | 18 | 2003,2020 | 12 | 207 |
| 24 | wheat, PL | 18 | 2003,2020 | 17 | 275 |
| 25 | wheat, PT | 17 | 2004,2020 | 5 | 85 |
| 26 | wheat, RO | 18 | 2003,2020 | 39 | 387 |
| 27 | wheat, SE | 17 | 2003,2020 | 17 | 246 |
| 28 | wheat, SK | 2 | 2017,2018 | 6 | 10 |
| 29 | wheat, US | 21 | 2003,2023 | 1937 | 21791 |

# 3  Related work

Table 5 shows a summary of related work to go along with *Section 2* of the main text.

Table 5: Overview of related work. The predictor acronyms are mapped as W-weather, SR- surface reflectance variables from remote sensing observations, VI-vegetation indices

| Reference | Bench mark | Coverage | Prediction unit | Time span | Data structure | Yield | W | SR | VI | Soil |
|---|---|---|---|---|---|---|---|---|---|---|
| (9) | ✓ | USA Argentina Brazil | county | 2005 - 2016 | image (histograms) | ✓ | ✓ | ✓ | | |
| (3) | | Nepal | district | 2016 - 2018 | image | ✓ | ✓ | ✓ | ✓ | ✓ |
| (10) | | China | province | 2016 - 2021 | image | ✓ | ✓ | ✓ | ✓ | |
| (4) | | Global | pixel | 1981 - 2016 | image | ✓ | ✓ | | ✓ | |
| (2) | | Germany | district | 1979 - 2021 | tabular | ✓ | | | | |
| (6) | | Netherlands | province | 1994 - 2018 | tabular | ✓ | | ✓ | ✓ | ✓ |
| (7) | | USA | county/ pixel | 2000 - 2018 | tabular image | ✓ | ✓ | ✓ | ✓ | ✓ |
| (5) | | USA | county | 1980 - 2018 | tabular | ✓ | ✓ | | | ✓ |
| (1) | | USA | pixel | 1999 - 2018 | image | ✓ | ✓ | | | |
| **CY-Bench** | ✓ | Mutil-country | sub-national | varies | tabular | ✓ | ✓ | | ✓ | ✓ |