# OpenReview forum: "CY-Bench: A comprehensive benchmark dataset for subnational crop yield forecasting"
_NeurIPS.cc/2024/Datasets_and_Benchmarks_Track — Submitted to NeurIPS 2024 Track Datasets and Benchmarks_

### Official Review · Reviewer_ZsHV · 2024-07-20
**A meanwhile benchmark**

**Rating:** 5
**Confidence:** 3
**Correctness:** Yes
**Clarity:** Yes

**Review:**

The authors provide detailed information on data sources, preprocessing steps, and evaluation metrics, which enhances the reproducibility and usability of the dataset. The paper is also written clearly and concisely, making it accessible to readers from various disciplines. The inclusion of tables, figures, and examples helps illustrate the dataset's features and the benchmarking process. The significance of CY-Bench lies in its potential to advance the state-of-the-art in crop yield forecasting by providing a comprehensive and standardized dataset. The benchmark facilitates the comparison of different modeling approaches, helping identify the most effective methods for improving food security and agricultural planning.

Pros

Comprehensive Dataset:  Covers a wide range of countries and crop types, providing extensive data for model training and evaluation.

Standardization:  Ensures consistent data preprocessing and evaluation, facilitating fair comparisons between different models.

Cons

No Differentiation Between Irrigated and Non-Irrigated Systems: This could lead to inaccuracies in yield forecasts for regions with mixed farming practices.

Absence of Weather Forecast Integration: Models cannot access future weather conditions beyond the lead time, potentially missing critical information that affects crop yields.

**Strengths:**

Comprehensive Data Integration: CY-Bench combines data from multiple sources, including weather, soil, and remote sensing, providing a rich dataset for model training and evaluation.

Standardization and Preprocessing: The standardized preprocessing steps ensure that the data is ready for analysis, reducing the effort required for data preparation.

Robust Benchmarking: The benchmark's design allows for robust and reproducible model evaluations, facilitating fair comparisons between different approaches.

**Additional Feedback:**

None

**Documentation:**

Yes

**Limitations:**

Yes

**Opportunities For Improvement:**

Absence of Weather Forecast Integration: Models cannot access future weather conditions beyond the lead time, potentially missing critical information that affects crop yields.

Differentiate Between Irrigated and Non-Irrigated Systems: The dataset does not distinguish between irrigated and non-irrigated farming systems, which can exhibit different responses to weather and soil conditions. This lack of differentiation may lead to inaccuracies in yield predictions for regions with mixed farming practices. By including this distinction, models can be better tailored to specific agricultural practices, resulting in more accurate and reliable yield forecasts. This would be particularly beneficial for regions where irrigation significantly impacts crop productivity.

Update Methodologies to Use Transformers Instead of LSTMs: The current methodology employs Long Short-Term Memory (LSTM) networks for time series forecasting. However, recent advancements in machine learning have shown that transformer models can outperform LSTMs in many time series forecasting tasks due to their ability to capture long-range dependencies more effectively. Updating the methodology to incorporate transformer models could lead to improved performance and scalability, making the forecasting models more accurate and efficient. This change would ensure that the benchmarking remains at the cutting edge of machine learning research, leveraging the latest advancements to achieve better results.

**Relation To Prior Work:**

Yes

**Summary And Contributions:**

The paper introduces CY-Bench, a comprehensive benchmark dataset designed for sub-national crop yield forecasting. The key contributions of the paper are:

Dataset Creation: CY-Bench includes crop yield data for maize and wheat across 42 countries, integrating various predictors such as weather, soil properties, and remote sensing indicators.

Standardization: The dataset standardizes data source selection, preprocessing, and spatio-temporal harmonization to facilitate consistent and comparable evaluations of machine learning models.

Multi-faceted Benchmarking: The dataset includes a wide range of predictors and supports various research efforts, including time series forecasting, domain adaptation, and online learning.

Public Access: The dataset and associated code are openly available, promoting further research in crop yield forecasting and related fields.

Interdisciplinary Collaboration: The dataset was curated by experts in agronomy, climate science, and machine learning, ensuring its relevance and applicability.

---

> ### Author Rebuttal · Authors · 2024-08-16
>
> We thank the reviewer for encouraging comments and suggestions. The reviewer raised three main weaknesses or opportunities for improvement. Our responses are as follows.
>
> > Absence of Weather Forecast Integration: Models cannot access future weather conditions beyond the lead time, potentially missing critical information that affects crop yields.
>
> Incorporating weather forecasts is potentially beneficial (Cunha et al, 2018), but outside the scope of our NeurIPS submission. However, we do plan to investigate the possibility of including them in a future version. There are multiple considerations that need to be discussed before including weather forecasts in CY-Bench: a) Observed weather data and forecast data may not come from the same source. b) Some variables, e.g. FPAR, NDVI, have no forecasts. We need a way to handle them. c) Weather forecasts from General Circulation Models (GCMs) typically have coarse spatial resolutions (50-400 km grid sizes) and often contain systematic errors or biases that must be adjusted. Bias correction and downscaling techniques must be applied using observed historical records of weather variables, such as precipitation and air temperature, to adjust the climate data and better represent local conditions. d) Using weather forecasts for yield prediction would also cause error/uncertainty propagation from the weather forecast models leading to an increase in overall uncertainty.
>
> > No Differentiation Between Irrigated and Non-Irrigated Systems: Irrigated and non-irrigated farming systems can exhibit different responses to weather and soil conditions. This could lead to inaccuracies in yield predictions for regions with mixed farming practices. By including this distinction, models can be better tailored to specific agricultural practices, resulting in more accurate and reliable yield forecasts. This would be particularly beneficial for regions where irrigation significantly impacts crop productivity.
>
> We appreciate the reviewer’s suggestion. However, this is a limitation from the publicly available data sources included in CY-Bench. Ideally, we would want to model them separately, but we are unable to do so because yield statistics are not always reported separately for irrigated versus non-irrigated systems. As a separate study, it is possible to investigate the value of modeling these separately for a few countries (e.g. Spain and the US), where yield statistics are reported separately.
>
> > Update Methodologies to Use Transformers Instead of LSTMs: The current methodology employs Long Short-Term Memory (LSTM) networks for time series forecasting. However, recent advancements in machine learning have shown that transformer models can outperform LSTMs in many time series forecasting tasks due to their ability to capture long-range dependencies more effectively. Updating the methodology to incorporate transformer models could lead to improved performance and scalability, making the forecasting models more accurate and efficient. This change would ensure that the benchmark remains at the cutting edge of machine learning research, leveraging the latest advancements to achieve better results.
>
> We agree with the reviewer. We intend to add the following models for the camera-ready version: a) Models forecasting residuals (from the average or yield trend). Residuals from the average are expected to account for location-specific differences, while residuals from the trend help capture year-to-year variability. This is a common modeling strategy used in the crop yield forecasting domain, b) 1-d convolutional networks as an alternative to recurrent networks, and c) Transformers as a state-of-the-art method for capturing long-term dependencies in time series data. Reviewer 1 (DYtA) also suggested adding ideas from recent studies.
>
> REFERENCES
>
> Cunha, R. L., Silva, B., & Netto, M. A. (2018, October). A scalable machine learning system for pre-season agriculture yield forecast. In 2018 IEEE 14th international conference on e-science (e-Science) (pp. 423-430). IEEE.

---

> > ### Author Response · Authors · 2024-08-29
> > **New models and updated results**
> >
> > Dear reviewer, here are some improvements we made based on your suggestions and our own timeline.
> >
> > 1. New models added:
> >     - Residual models: For each machine learning model, we have implemented a version that subtracts the trend computed using a linear trend model and forecasts the residuals. The final predictions add the trend back, producing yield forecasts. Some of the residual models perform better than their “normal” yield forecasting counterparts.
> >
> >    - InceptionTime model: 1-D CNN based model. The implementation is based on [1]. Results are listed in `tables_aug2024.md` in [GitHub](https://github.com/BigDataWUR/AgML-CY-Bench/tree/main/results_baselines/tables).
> >
> >     - Transformer model: The first version is based on [1]. Preliminary results in terms of normalized RMSE obtained from maize France are comparable to those obtained with the InceptionTime and LSTM models (i.e. Transformer: 17.3 vs InceptionTime: 19.1, LSTM: 15.8). We will continue to adapt the implementation to the specifics of our task.
> >
> > 2. We have updated the results after addressing some issues in our data preprocessing pipelines, in particular aligning the input data to the crop growing season. The results tables are named `tables_original.md` (submitted with the paper in June), `tables_june2024.md` (rerun in June), `tables_aug2024.md` (updated in August). In the updated results for maize, machine learning models performed better than the Naive baseline (i.e. have lower normalized RMSE) in 26 out of 38 cases (compared to 11 cases in the June 2024 run). For wheat, machine learning models performed better in 11 out of 28 cases (compared to 5 cases in the June 2024 run). We also note that these are results for middle-of-season forecasts. Results generally improve as the lead time is closer to the end of the season.
> >
> > If you feel that we have addressed your concerns, please consider updating the score accordingly.
> >
> >
> > REFERENCES
> >
> > [1] Oguiza, I. tsai-a state-of-the-art deep learning library for time series and sequential data. GitHub (2023). URL https://github.com/timeseriesAI/tsai.

---

### Official Review · Reviewer_nRmX · 2024-07-24
**Informative Paper and High-Quality Dataset**

**Rating:** 8
**Confidence:** 4
**Correctness:** The claims made in the submission are…
**Clarity:** The paper is well-written and clear.

**Review:**

This is a very solid work. The pros and cons are outlined below:

Pros:
1. Multidisciplinary Cooperation: The dataset and benchmark involve scientists and experts from agriculture, climate, and machine learning fields, showcasing well-coordinated and knowledgeable work.
2. High-Quality Dataset: The dataset is aggregated from multiple sources and aligns well with different types of datasets, showing good impact across multiple research domains.
3. Good Summary of Limitations and Future Work: The limitations and future work section is well-written and provides a clear direction for further research.
4. Good Writing: The paper is well-written and easy to follow.

Cons:
1. Inadequate Comparison with Existing Datasets: CY-Bench is not compared with existing datasets listed in related work.
2. Lacks Details: Some data and feature selection details are missing.
3. Lacks Ablation Study: No ablation study on feature selection is provided.

**Strengths:**

1. Multidisciplinary Cooperation: The collaboration between experts from various domains enhances the quality and applicability of the dataset and benchmark.
2. High-Quality Dataset: The aggregation of data from multiple sources ensures the dataset’s relevance and utility across different research areas.
3. Good Limitations and Future Work Summary: The section provides a comprehensive overview of the limitations and future directions, guiding further improvements.
4. Good Writing: The paper is clear, concise, and easy to follow.

**Additional Feedback:**

I am curious about the following questions:
1. Can the dataset be easily extended to other types of crops or countries? What challenges would this entail?
2. Does the dataset consider fine-grained temporal resolution?
3. Does the benchmark address extreme yield data?
4. What is the reason behind selecting the mid-season for peak model performance? The paper mentions that peak performance typically reaches around the mid-point of the growing season, but further explanation would be beneficial.

**Documentation:**

The documentation is well-prepared.

**Ethics:**

There are no ethical concerns with the dataset or the paper.

**Limitations:**

The limitations are well-addressed.

**Opportunities For Improvement:**

1. Adding Comparisons with Existing Datasets: Comparative experiments with existing datasets can clarify the contribution of CY-Bench. Metrics such as duration, temporal, and spatial resolution should be included, even if other datasets cover only one country.
2. Providing More Details on Data Source and Feature Selection: More information is needed on how different data sources and features were selected. For example, why was AgERA5 chosen for the meteorological dataset over other global datasets? What specific reasons are for selecting features like temperature, precipitation, and solar radiation?
3. Including an Ablation Study: An ablation study to demonstrate the rationale behind the dataset and feature selection would enhance the overall quality of the dataset. This effort could provide deeper insights into the dataset’s construction and its impact on model performance.

**Relation To Prior Work:**

Related works are discussed well.

**Summary And Contributions:**

CY-Bench is a comprehensive dataset and benchmark for crop yield forecasting. Scientists from various domains collaborated on CY-Bench to standardize data source selection, preprocessing, spatio-temporal harmonization, and machine learning model evaluation. The dataset covers 29/42 countries for Wheat/Maize, making it significant in size. The dataset and benchmark have potential implications for global food security.

---

> ### Author Rebuttal · Authors · 2024-08-16
>
> We thank the reviewer for their positive and constructive feedback. Point-by-point responses are provided below.
>
> ## Opportunities for improvement
> > Inadequate Comparison with Existing Datasets: CY-Bench is not compared with existing datasets listed in related work. Comparative experiments with existing datasets can clarify the contribution of CY-Bench. Metrics such as duration, temporal, and spatial resolution should be included, even if other datasets cover only one country.
>
> The supplement (Section 3. Related Work) includes a table that compares CY-Bench with existing datasets considering geographic coverage, prediction unit or spatial resolution of the target variable, time span, data structure, and predictors. To make the comparison more clear, we will: a) modify the table caption to “Comparison of CY-Bench with existing datasets” from “Overview of Related Work”, and b) add an explicit reference to the table in the main text.
>
> > Providing More Details on Data Source and Feature Selection: More information is needed on how different data sources and features were selected. For example, why was AgERA5 chosen for the meteorological dataset over other global datasets? What specific reasons are for selecting features like temperature, precipitation, and solar radiation?
>
> We have documented the selection of data sources (e.g. weather, remote sensing, soil moisture, soil properties, crop calendars) in [GitHub](https://github.com/BigDataWUR/AgML-CY-Bench/blob/main/data_preparation/DATA-SOURCES-SELECTION.md) and provided a reference to this in the supplement. Guiding principles for selection of data sources were global coverage, public access, regular updates (except for static data) and near real-time availability. Input variables were selected after detailed discussions about their pros and cons based on their importance for crop growth and development. We will add a table in the supplementary information justifying our selection of input variables.
>
> > Including an Ablation Study: An ablation study to demonstrate the rationale behind the dataset and feature selection would enhance the overall quality of the dataset. This effort could provide deeper insights into the dataset’s construction and its impact on model performance.
>
> We agree with the suggestion of including an ablation study comparing the relative model performance improvements provided by different inputs. This would provide useful context for researchers intending to work with the dataset, and insights gained can be usefully compared with those from previous studies that have assessed this for specific crops and countries. For the camera-ready version, we therefore intend to include an ablation study which compares model performance using varying combinations of weather features, soil data and remote-sensing derived vegetation indices (FPAR, NDVI). We have created [an issue in GitHub](https://github.com/BigDataWUR/AgML-CY-Bench/issues/303) to track this.
>
>
> ## Additional Feedback
> > Can the dataset be easily extended to other types of crops or countries? What challenges would this entail?
>
> Yes, the dataset can be easily extended to other crops and countries. The main challenge would be obtaining yield statistics, crop calendars, and crop masks. Given this information, our pipeline can be used to prepare data for additional crops or countries.
>
> > Does the dataset consider fine-grained temporal resolution?
>
> The input data used retain their temporal resolution from the original data source. Model contributors can decide whether to aggregate input data or not. For scikit-learn baselines (Ridge and Random Forest), monthly aggregation is done during feature extraction. For LSTM, we aggregate all time series inputs to the same temporal resolution which allows all data points to have the same number of time steps and therefore fixed input dimension.
>
> > Does the benchmark address extreme yield data?
>
> Extreme yield data is captured when the timeframe of the original yield statistics cover the extreme events. For model evaluation, we selected a leave-one-year-out (LOYO) scheme mainly to gauge performance on extreme yields. Any other split may or may not guarantee a test set with extreme yields in it.
>
> > What is the reason behind selecting the mid-season for peak model performance? The paper mentions that peak performance typically reaches around the midpoint of the growing season, but further explanation would be beneficial.
>
> We would like to thank the reviewer for raising this. The statement should have been about the most important period for plants, rather than about peak model performance. Model performance may or may not peak at mid-season. For many crops, the reproductive period (flowering and grain filling) is the most climate-sensitive portion of the growing season. This is what we mean by mid-season. After grain filling during senescence, the crop is less sensitive to climate anomalies. Before mid-season during the vegetative phase, the plant is also less sensitive to climate anomalies. However, we don’t want to imply that the mid-season is the only time of climate sensitivity. For example emergence may be another climate sensitive portion of the season.
>
> Quarter-of-season (relative to harvest), when plants reach physiological maturity, ideally would provide better performance. But, the forecasts made during mid-season (during grain filling) usually provide a reasonable estimate of yield while also providing an opportunity for management adjustments late in the season that quarter-of-season would not provide. We will include results from both middle-of-season and quarter-of-season (relative to harvest) in the camera-ready version.

---

### Official Review · Reviewer_DYtA · 2024-08-07
**Countries are comprehensively covered**

**Rating:** 5
**Confidence:** 4

**Review:**

### Motivation & Contributions / Quality
Predicting crop yield is a highly relevant task to society for humanitarian and economic reasons. Previously published machine learning research dedicated to crop yield often lacks publicly available datasets. There exist multiple datasets and benchmarks for crop yield forecasts, focusing on different crops and using different covariates. CY-Bench mainly differentiates itself by including more countries than previous work. However, some previous works are richer in the number of crops [1] included and have informative covariates [1 ,2, 3] (e.g., Satellite images in CropNet [2], management data in [3]).

I agree with the authors that including more countries is useful. Especially, developing countries have been underrepresented in previously published machine learning-based crop yield prediction, which mostly focuses on the US and EU states.

In their experiments, the authors did not include any recent contributions for crop yield forecasting and instead exclusively focused on traditional machine learning methods (Ridge regression, Random Forest, LSTM).

### Clarity
The shared code and documentation are very detailed and intuitive to use.
It is hard to understand the exact training task of the benchmark, which relates to the authors' decision to omit a formal (mathematical) problem formulation.

### Significance & Originality
CY-Bench contributes to the machine learning community by providing a crop-yield forecasting benchmark that includes a higher number of countries than previously published datasets. Additionally, CY-Bench lowers the entry barrier into machine learning-based crop yield forecasting with excellent and well-documented code.

[1] Paudel, Dilli, et al. "Machine learning for large-scale crop yield forecasting." Agricultural Systems 187 (2021): 103016.

[2] Lin, Fudong, et al. "CropNet: An Open Large-Scale Dataset with Multiple Modalities for Climate Change-aware Crop Yield Predictions."

[3] Khaki, Saeed, Lizhi Wang, and Sotirios V. Archontoulis. "A CNN-RNN framework for crop yield prediction." Frontiers in Plant Science 10 (2020): 1750.

**Strengths:**

(S1) CY-Bench is a comprehensive benchmark dataset that includes crop yield prediction tasks for many countries.

(S2) The related code is well-documented, and CY-Bench is easy to use.

(S3) The composition and preprocessing of the data are well-documented in the paper.

**Additional Feedback:**

While the collection of crop yield data is sufficiently comprehensive, the tasks contained in the benchmark might have insufficient training samples, which could be the reason for machine learning models failing. The small data sizes of certain tasks is an artifact of dividing tasks by country. The authors provide no evidence that it is necessary to divide tasks by countries. Geographical differences can be covered by the covariate features (weather, soil) and also exist within nations, especially in large ones. A solution for this could be to divide into tasks into groups of countries instead of single countries.

**Clarity:**

In general the paper is well written and easy to follow. However, it is not trivial to understand the exact training task of the benchmark, due to the lack of a formal problem definition.

On p. 6 there is are double parenthesis "((51))"
The plots on p. 7 are hard to read, when the paper is printed. Maybe one could increase the font-size.

**Correctness:**

The submission claims to contribute a comprehensive sub-national crop yield forecasting benchmark that covers maize and wheat, that will facilitate the developement of crop yield forecasting models in the future. However, there exists benchmarks that train models with more information [1,2,3], that would be available in a real-world scenario.  I think it can be problematic to use CY-Bench to evaluate future research, and instead focus on experiments with more covariates. Simplicity is a less important feature for such a relevant task.

[1] Paudel, Dilli, et al. "Machine learning for large-scale crop yield forecasting." Agricultural Systems 187 (2021): 103016.

[2] Lin, Fudong, et al. "CropNet: An Open Large-Scale Dataset with Multiple Modalities for Climate Change-aware Crop Yield Predictions."

[3] Khaki, Saeed, Lizhi Wang, and Sotirios V. Archontoulis. "A CNN-RNN framework for crop yield prediction." Frontiers in Plant Science 10 (2020): 1750.

**Documentation:**

Yes, the dataset and code are easy to find and it is trivial to rerun the experiments. Furthermore the authors provide easy-to-follow instructions on how to run a new model on CYBench.

**Ethics:**

I do not have any ethical concerns

**Limitations:**

I see no negative societal impact in this submission. Instead crop-yield forecasting has the potential to have a vastly positive societal impact.
The authores dedicated half a page to discuss limitations including incomprehensive predictors  and the assumptions that all years are indepented (therefore ignoring e.g. climate trends).

**Opportunities For Improvement:**

(W1) Many tasks have small data, which is problably part of the explanation that machine learning models fail to outperform a naive baseline.

(W2) Lack of a formal problem formulation.

(W3) More recent models that can actually outperform the naive baseline could be included in the experiments / baseline models

(W4) CY-Bench misses important features that were already used in previous works (e.g., satellite images in CropNet).

**Relation To Prior Work:**

The authors mention multiple previously published crop forecasting benchmarks, highlighting SustainBench[1] and CropNet[2]. However, in my opinion, [3] is the most similar to CY-Bench. I cannot comprehend why the authors mention this work as not explicitly tailored for crop yield forecasting.

[1] Yeh, Christopher, et al. "Sustainbench: Benchmarks for monitoring the sustainable development goals with machine learning." arXiv preprint arXiv:2111.04724 (2021).

[2] Lin, Fudong, et al. "CropNet: An Open Large-Scale Dataset with Multiple Modalities for Climate Change-aware Crop Yield Predictions."

[3] Paudel, Dilli, et al. "Machine learning for large-scale crop yield forecasting." Agricultural Systems 187 (2021): 103016.

**Summary And Contributions:**

The proposed work provides an extensive collection of sub-national crop-yield data combined with relevant covariates, such as information about weather and soil.

Based on the collected data, the authors propose CY-Bench, a benchmark for crop-yield forecasting that consists of mid-season yield forecasts for wheat and corn in a multitude of countries.

The task is to estimate the harvest yield of a sub-national region at the end of the harvest season based on predictors (weather, soil data) given until mid-harvest season. For training instances, they use the sub-national crop yields of other years from all sub-national regions in the respective country.

In their experiments, the authors show that, in CY-Bench, traditional machine learning baselines fail to outperform a naive baseline in many cases.

---

> ### Author Rebuttal · Authors · 2024-08-16
>
> We thank the reviewer for valuable suggestions and constructive feedback.
>
> > Previous work covers more crops.
>
> We note that [6] and [3] are not public benchmarks, but publications that share sample data. [5] covers only one country (US) for 6 years. While CY-Bench covers only maize and wheat, they are the most important crops in terms of global food trade [1] and diets in the majority of countries [4]. CY-Bench coverage exceeds any previously published benchmark for the task. Other crops are not included because they don’t have globally available crop masks and crop calendars.
>
> > There are benchmarks that train models with more informative covariates, including satellite images [5] and management data [3], that would be available in a real-world setting. It will be problematic to use CY-Bench to evaluate future research, and instead focus on experiments with more covariates. Simplicity is a less important feature for such a relevant task.
>
> Our selection of covariates is guided by global availability of well-maintained data sources. [3] use cumulative weekly percentages of planted fields, which may not be available globally. CY-Bench does not train with satellite imagery, but includes vegetation indices from remote sensing (NDVI – the most frequently-used index for crop yield forecasting [8], and FPAR), that are strongly correlated with yields [2].The utility of high resolution (10-60m) satellite data for crop yield forecasting at subnational level is not very clear. The time series is also not very long (e.g, Sentinel available from 2014/15). Still, we agree that including satellite imagery would be a promising direction for future work.
>
> > More recent models that can actually outperform the naive baseline could be included in the experiments / baseline models. In their experiments, the authors did not include any recent contributions for crop yield forecasting and instead exclusively focused on traditional machine learning methods (Ridge regression, Random Forest, LSTM).
>
> We intend to add the following models: a) Models forecasting residuals (from the average or yield trend) to account for location-specific differences and year-to-year variability. This is a common modeling strategy used in crop yield forecasting, b) 1-d convolutional networks as an alternative to recurrent networks, and c) Transformers as a state-of-the-art method for capturing long-term dependencies in time series data.
>
> > It is hard to understand the exact training task of the benchmark, which relates to the authors' decision to omit a formal (mathematical) problem formulation.
>
> CY-Bench is designed to evaluate model performance for in-season crop yield forecasting. Forecasts are made at multiple time points from start of season ($SOS$) to end of season ($EOS$), based on a lead time relative to EOS. We report forecasts for middle-of-season ($(EOS - SOS)/2$). Other options are quarter-of-season ($(EOS - SOS)/4$), or $n$-days before harvest.
>
> The input data consists of time series inputs (weather, soil moisture and vegetation indices) and static inputs (soil properties). Let $x_t$ represent a vector of time series input at time $t$, which ranges from $SOS$ to the inference point $T$. Time series data up to $T$ is represented as $X_{SOS:T} = (x_{SOS}, x_{SOS+1}, …, x_T)$ and static inputs as $z$. Each training or testing sample corresponds to a specific region-season pair $(r, s)$. For each training sample $i = (r, s)$, the input consists of $X_{SOS:T}^{(i)}$ and $z^{(i)}$. The target is the end-of-season yield $Y^{(i)}$. The objective is to learn a function f such that $Y^{(i)} = f(X_{SOS:T}^{(i)}, z^{(i)};\theta) + \epsilon^{(i)}$, where $\theta$ represents the model parameters, and $\epsilon^{(i)}$ is the error term.
>
> During testing, the model gets $X_{SOS:T}^{(j)}$ and $z^{(j)}$ for a new sample $j = (r', s')$. The model then forecasts the end-of-season yield $\hat{Y}^{(j)} = f(X_{SOS:T}^{(j)}, z^{(j)}; \hat{\theta})$, where $\hat{\theta}$ are the model parameters learned during training. Model performance is evaluated by comparing yield forecasts $\hat{Y}^{(j)}$ with reported yields $Y^{(j)}$.
>
> > On p. 6 there is are double parenthesis "((51))". The plots on p. 7 are hard to read, when the paper is printed. Maybe one could increase the font-size.
>
> We will fix the typo and update all figures to make them more readable.
>
> > Many tasks have small data, which is probably part of the explanation that machine learning models fail to outperform a naive baseline. The small data sizes of certain tasks is an artifact of dividing tasks by country. The authors provide no evidence that it is necessary to divide tasks by countries. Geographical differences can be covered by the covariate features (weather, soil) and also exist within nations, especially in large ones. A solution for this could be to divide tasks into groups of countries instead of single countries.
>
> Although our baseline models are trained with per-country data, this is not a hard requirement. The modeler is free to use samples from other countries. We show results per country because determining similarity among countries is not straightforward due to differences in sizes of admin regions, farm management, technology adoption, agricultural policies and protocols for crop statistics collection. One approach would be to group subnational units based on agro-environmental similarities. [7] have shown that this may not always provide added benefit. We will investigate grouping countries when training large neural networks.
>
> > In my opinion, [6] is most similar to CY-Bench. I cannot comprehend why the authors mention this work as not explicitly tailored for crop yield forecasting.
>
> We note that some cited studies ([6], [3]) publish sample data, but are not public benchmarks. Therefore, our comment was about benchmarking rather than crop yield forecasting. Even then, we thank the reviewer for pointing this out. We will update the sentence to clarify our message.

---

> > ### Author Response · Authors · 2024-08-16
> > **References to DYtA rebuttal**
> >
> > [1] D’Odorico, Paolo, et al. “Feeding humanity through global food trade.” Earth’s Future 2 (2014): 458-469
> >
> > [2] Johnson, D. M. “A comprehensive assessment of the correlations between field crop yields and commonly used MODIS products.” International journal of applied earth observation and geoinformation (2016): 52, 65-81.
> >
> > [3] Khaki, Saeed, Lizhi Wang, and Sotirios V. Archontoulis. "A CNN-RNN framework for crop yield prediction." Frontiers in Plant Science 10 (2020): 1750.
> >
> > [4] Khoury, Colin, et al. “Increasing homogeneity in global food supplies and the implications for food security.” Proceedings of the National Academy of Sciences 111 (2014): 4001-4006
> >
> > [5] Lin, Fudong, et al. "CropNet: An Open Large-Scale Dataset with Multiple Modalities for Climate Change-aware Crop Yield Predictions.”
> >
> > [6] Paudel, Dilli, et al. "Machine learning for large-scale crop yield forecasting." Agricultural Systems 187 (2021): 103016.
> >
> > [7] Paudel, Dilli, et al. "Machine learning for regional crop yield forecasting in Europe." Field Crops Research 276 (2022): 108377.
> >
> > [8] Schauberger, Bernhard, et al. “A systematic review of local to regional yield forecasting approaches and frequently used data resources.” European Journal of Agronomy 120 (2020): 126153.
> >
> > [9] Yeh, Christopher, et al. "SustainBench: Benchmarks for monitoring the sustainable development goals with machine learning." arXiv preprint arXiv:2111.04724 (2021).

---

> > ### Comment · Reviewer_DYtA · 2024-08-28
> >
> > Dear Authors,
> >
> > Thank you for your detailed and comprehensive response. I appreciate the effort you've put into addressing my previous concerns.
> >
> > However, I still have some concerns regarding the novelty and whether the dataset contains sufficient richness to support the findings.
> >
> > That said, I recognize the merits of your work and will raise my score to a 5.

---

> > ### Author Response · Authors · 2024-08-29
> > **New models and updated results**
> >
> > Dear reviewer, We would like to thank you for your response. Here are some improvements we made based on your suggestions and our own timeline.
> >
> > 1. New models added:
> >     - Residual models: For each machine learning model, we have implemented a version that subtracts the trend computed using a linear trend model and forecasts the residuals. The final predictions add the trend back, producing yield forecasts. Some of the residual models perform better than their “normal” yield forecasting counterparts. Results are listed in `tables_aug2024.md` in [GitHub](https://github.com/BigDataWUR/AgML-CY-Bench/tree/main/results_baselines/tables).
> >
> >    - InceptionTime model: 1-D CNN based model. The implementation is based on [1]. Results are listed in `tables_aug2024.md` in [GitHub](https://github.com/BigDataWUR/AgML-CY-Bench/tree/main/results_baselines/tables).
> >
> >     - Transformer model: The first version is based on [1]. Preliminary results in terms of normalized RMSE obtained from maize France are comparable to those obtained with the InceptionTime and LSTM models (i.e. Transformer: 17.3 vs InceptionTime: 19.1, LSTM: 15.8). We will continue to adapt the implementation to the specifics of our task.
> >
> > 2. We have investigated the idea of grouping countries.
> >
> >     - We combined data from BE, DE, NL, DK in Europe. The selection is based on geographical closeness and agro-environmental similarities. We see that DK (with the smallest data size of 22) benefits a little bit from the increased data size (the best normalized RMSE is 16.69% compared to 18.83% without grouping). For other countries included in the group, there is no noticeable advantage. For more details, please check `tables_group.md` and `tables_aug2024.md` in [GitHub](https://github.com/BigDataWUR/AgML-CY-Bench/tree/main/results_baselines/tables).
> >
> >    - Similarly, we combined data from AO, ZM, MZ and MW in Africa. We see that the results for MW (with the smallest data size of 16) does not benefit from the increased data size. For other countries, the performance does not change much either. See tables mentioned earlier for details.
> >
> >     - Overall, we see that gains are small. There are a number of issues we see when grouping countries (in addition to those mentioned in our previous response). (i) The crop (maize or wheat) can mean something different in different countries. See [Deciphering crop names in our overview](https://github.com/BigDataWUR/AgML-CY-Bench?tab=readme-ov-file#overview). (ii) Our evaluation method builds a model for each test year. Training data comes from the remaining years. In the case of small data sizes, countries often have missing data for several years. Therefore, data for different countries may not come from the same years. (iii) Neural networks require fixed input dimension (or number of time steps) for time series data. When combining data from multiple countries, there can be mismatches in crop season length. For example, country A and B may have crop seasons of lengths 100 and 200 days. Forcing the time series data from different countries to have the same number of time steps leads to information loss for country B as data from the start of the season are removed to ensure the same input dimension.
> >
> > 3. We have updated the results after addressing some issues in our data preprocessing pipelines, in particular aligning the input data to the crop growing season. The results tables are named `tables_original.md` (submitted with the paper in June), `tables_june2024.md` (rerun in June), `tables_aug2024.md` (updated in August). In the updated results for maize, machine learning models performed better than the Naive baseline (i.e. have lower normalized RMSE) in 26 out of 38 cases (compared to 11 cases in the June 2024 run). For wheat, machine learning models performed better in 11 out of 28 cases (compared to 5 cases in the June 2024 run). We also note that these are results for middle-of-season forecasts. Results generally improve as the lead time is closer to the end of the season.
> > 4. We agree with the reviewer’s concerns about the novelty of models and richness of data. We are actively improving the suite of models included in the benchmark. For input data, we are constrained by global availability (e.g. for farm management data). We have previously addressed the comment about the use of raw satellite images.
> >
> > If you feel that we have addressed your concerns, please consider updating the score accordingly.
> >
> >
> > REFERENCES
> >
> > [1] Oguiza, I. tsai-a state-of-the-art deep learning library for time series and sequential data. GitHub (2023). URL https://github.com/timeseriesAI/tsai.

---

### Decision · Program_Chairs · 2024-09-26

**Decision:**

Reject

**Comment:**

The reviewers see several strengths in the paper:
- s1. interesting dataset on crop yield prediction covering many countries.
- s2. data integration from several sources
- s3. preprocessing standardized and well documented.

But they also discussed several weaknesses:
- w1. many tasks have only small data and thus may be not so interesting for ML models.
- w2. some important data aspects are missing: weather forecasts, maybe satellite images.
- w3. more recent models (e.g., transformers instead of LSTMs) should be added.
- w4. the ML tasks could be formulated more clearly and explicitly.

The authors addressed w3 "more recent models" during the rebuttal.
But given the other weaknesses, overall I do not recommend to accept
the paper in its current form.